# The absence of orthostatic heart rate increase is associated with cognitive impairment in Parkinson's disease

**Ryota Tanaka**[1,2]*, **Kazuo Yamashiro**[3], **Takashi Ogawa**[2], **Genko Oyama**[2], **Kenya Nishioka**[2], **Atsushi Umemura**[4], **Yasushi Shimo**[5], **Nobutaka Hattori**[2]*

**1** Stroke Center, Jichi Medical University Hospital, Division of Neurology, Department of Medicine, Jichi Medical University, Tochigi, Japan, **2** Department of Neurology, Juntendo University, Tokyo, Japan, **3** Department of Neurology, Juntendo University Urayasu Hospital, Chiba, Japan, **4** Department of Neurosurgery, Juntendo University, Tokyo, Japan, **5** Department of Neurology, Juntendo University Nerima Hospital, Tokyo, Japan

\* rtanaka@jichi.ac.jp (RT); n_hattori@juntendo.ac.jp (NH)

**Data Availability Statement:** All relevant data are within the paper.

**Funding:** The author(s) received no specific funding for this work.

## Abstract

Orthostatic hypotension (OH) frequently accompanies autonomic dysfunction and is an important risk factor for cognitive impairment in Parkinson's disease (PD). While OH is usually diagnosed based on an orthostatic blood pressure drop, the association between the heart rate response and cognitive impairment remains unclear. We retrospectively analyzed 143 cases of clinically diagnosed PD to determine the association between the absence of a heart rate response and cognitive impairment in PD with OH. Among the patients with OH, neurogenic OH was diagnosed in cases without a heart rate increase, while all other patients were diagnosed with non-neurogenic OH. Dementia was found in 23 of 143 PD cases (16.1%) in this cohort. The presence of OH was an independent risk factor for dementia in PD in addition to the disease severity, years of education and beta-blockers use. Neurogenic OH was significantly associated with dementia compared to the no OH group (hazard ratio [HR] 7.3, 95% confidence interval [CI] 2.2–24.6, P<0.01), an association that was preserved after adjusting for age, gender and other covariant factors. However, no such association was observed for non-neurogenic OH (HR 2.9, 95%CI 0.8–10.9, P = 0.12). While the cognitive impairment was significantly worse in the neurogenic OH group than the no-OH group, the groups were otherwise similar. The blood pressure decrease was significantly lower in both OH groups than in the no-OH group, despite no significant differences between the OH groups. Our finding showed that OH without a heart rate response was an important predictor of cognitive impairment in PD.

## Introduction

Parkinson's disease (PD) is a neurodegenerative disorder with a middle-age onset, and it manifests as progressive motor symptoms, including bradykinesia, muscular rigidity, tremor at rest, and postural or gait disturbance [1, 2]. Non-motor symptoms, such as cognitive decline and autonomic dysfunction, are important factors that also affect the prognosis of PD [3, 4].

**Competing interests:** The authors have read the journal's policy and have the following competing interests: RT received honoraria for work unrelated to this study from Takeda Pharmaceutical Co.,Ltd., Nippon Boehringer Ingelheim,Co.,Ltd, Dai-Nippon Sumitomo Pharma Co.,Ltd., Bayer Yakuhin, Ltd, Otsuka Pharmaceutical, Co.,Ltd, Pfizer Japan Inc., DAIICHI SANKYO Co.,Ltd., Eisai Co.,Ltd., Bristol-Myers Squibb Co., Stryker Japan K.K., CSL Behring K.K,. and Kowa Co.,Ltd. KY received honoraria for work unrelated to this study from Pfizer Inc., Takeda Pharmaceutical Co.,Ltd., Dai-Nippon Sumitomo Pharma Co.,Ltd., Bayer Yakuhin, Ltd., Otsuka Pharmaceutical, Co.,Ltd., DAIICHI SANKYO Co.,Ltd., and Novartis Pharma K.K. KN received a research grant for work unrelated to this study from Biogen Japan, Ltd.GO received honoraria for work unrelated to this study from Medtronic, Boston Scientific, Otsuka Pharmaceutical, Sumitomo Dainippon Pharma, Novartis Pharma, MSD, Nihon Medi-Physics, FP Pharmaceutical Corporation, Kyowa Hakko Kirin, and AbbVie, Inc. AU received a research grant from Novartis Pharma K.K, consultancy fees from Boston Scientific Japan, and honoraria from Medtronic Japan Inc., Novartis Pharma K.K, Kyowa Hakko-Kirin Co.,Ltd., Takeda Pharmaceutical Co.,Ltd. and AbbVie, Inc. all for work unrelated to this study. YS received honoraria for work unrelated to this study from Medtronic, Boston Scientific, Otsuka Pharmaceutical, Takeda Pharmaceutical CO, Sumitomo Dainippon Pharma, Novartis Pharma, MSD, FP Pharmaceutical Corporation, Kyowa Hakko Kirin, and AbbVie, Inc. NH received grants from AbbVie GK, FP Pharma. Co, Dai-Nippon Sumitomo Pharma Co.,Ltd, Eisai Co.,Ltd. and honoraria from Kyowa Kirin Co., Ltd., Dai-Nippon Sumitomo Pharma Co., Ltd, Takeda Pharmaceutical Co., Ltd., Otsuka Pharmaceutical, Co., Ltd, AbbVie GK, Eisai Co., Ltd., FP Pharma. Co, Mitsubishi Tanabe Pharma Co., Otsuka Pharmaceutical, Novartis Pharma, MSD, Nihon Medi-Physics, Bristol-Myers Squibb Company, Ono Pharmaceutical Co., Ltd., EA Pharma Co., Ltd, Asahi Kasei Medical Co., Ltd., and Chugai Pharma Manufacturing Co., Ltd., all for work unrelated to this study. NH also received honoraria from Biogen Japan, Ltd. for subcontracting (trial cases) and has the Equity stock (8%) of PARKINSON Laboratories Co. Ltd. This does not alter our adherence to PLOS ONE policies on sharing data and materials. There are no patents, products in development or marketed products associated with this research to declare.

Orthostatic hypotension (OH) is one of the most frequently observed examples of autonomic dysfunction in PD [5]. OH is usually classified into sub-types of neurogenic OH, which shows decrease in the orthostatic blood pressure (BP) without a compensatory heart rate increase, and non-neurogenic OH, which does show a heart rate increase. Lewy body pathology associated with cardiovascular autonomic dysfunction causes neurogenic OH in PD [5], while non-neurogenic OH is usually caused by hypovolemia and cardiac pump failure.

The existence of OH has been associated with falling, and the prodrug to norepinephrine has been shown to reduce the risk of falling in cases of PD [6, 7]. Furthermore, the coexistence of OH and other autonomic dysfunctions in PD was found to be associated with a poorer survival rate over long-term observation [8, 9]. Cognitive decline and dementia are also important factors affecting the prognosis of PD [10, 11]. Approximately 30% of PD patients develop dementia, and OH has been consider an independent risk factor for cognitive decline, along with one's age, an older age at onset, akinetic-rigid subtypes, and non-motor symptoms such as visual hallucination and, rapid eye movement sleep behavior disorders in PD [12].

While OH is usually diagnosed based on a decrease in BP within 3 minutes after rising from a supine position, the absence of a heart rate increase is an important response for discriminating neurogenic OH from non-neurogenic OH [13]. However, few studies have assessed the association between the presence or absence of heart rate response and cognitive decline in cases of PD with OH.

In the present study, we assessed whether or not the absence of heart rate increase was a risk factor for dementia in PD patients with OH.

## Material and methods

We used a retrospective cohort to analyze the association between OH and dementia in PD. We conducted a retrospective review of 172 patients with PD admitted to Juntendo University Hospital for a diagnostic assessment, drug adjustment, or evaluation for deep-brain stimulation between January 2014 and October 2017. We excluded patients with PD admitted for the treatment of acute illnesses, such as acute infection and ileus, and also excluded patients with PD who had congestive heart failure and diabetes mellitus. The diagnosis of PD was made according to the UK Brain Bank criteria [1].

Of the 172 participants, 20 were excluded due to the absence of an OH evaluation or cognitive assessment. We also excluded nine patients who had already received anti-hypotensive medication. We collected the baseline characteristic of patients, such as the age, duration of disease, Hoehn-Yahr stage (H-Y), body mass index (BMI), history of hypertension, stroke, coronary artery disease, peripheral artery disease, and anti-hypertensive medication such as angiotensin converting enzyme inhibitors (ACE-Is). angiotensin II receptor blockers (ARBs), calcium channel blocker (CCBs), beta blockers, and diuretics using medical records. We also calculated the levodopa equivalent daily dose (LEDD) for each participant.

We obtained oral informed consent from the participants and provided patients with the opportunity to opt out. The study protocol was approved by the ethics committee of Juntendo University Hospital.

### OH and supine hypertension (SH)

After at least 15 minutes resting in the supine position, the BP was measured, using an electronic sphygmomanometer (ES-H55; Terumo, Tokyo, Japan). The first measurement was taken while the patient remained supine, followed by a BP assessment in a standing position. OH was defined as a 20-mmHg drop in systolic BP and/or a 10-mmHg drop in diastolic BP within the first 3 minutes after standing.

The baseline supine and the lowest orthostatic values for blood pressure were recorded. Furthermore, we determined the maximum increase in heart rate within 3 minutes after postural change. If the patient's heart rate (HR) increase was < 15 beats per minute, we diagnosed them with neurogenic OH, whereas if the patient's HR increase was ≥ 15 beats per minute, we diagnosed them with non-neurogenic OH.

SH was defined as a systolic BP of ≥140 mmHg or a diastolic BP of ≥90 mmHg, when in the supine position.

### The cognitive assessment and diagnosis of dementia

The cognitive function was assessed using the Mini-Mental State Examination (MMSE), the Hasegawa dementia scale-revised (HDS-R), and the Montreal Cognitive Assessment (MoCA). We enrolled PD with dementia (PDD) patients but not dementia with Lewy bodies (DLB) patients based on the "one-year rule" [14], and the diagnosis of PDD was based on the diagnostic criteria from the Movement Disorder Society Task Force [15].

### Detection of cerebrovascular lesions

Brain magnetic resonance imaging (MRI) was performed using a 1.5-T MR system (Vistart RX; Toshiba, Japan). The whole brain was scanned at a slice thickness of 5.5mm with an inter-slice gap of 1mm; 20 axial images were obtained. The imaging protocol consisted of axial fluid-attenuated inversion recovery (FALIR) images for small vessel disease. Deep white matter hyperintensity (DWMH), was also assessed with MRI using semiquantitative visual scales [16].

### Statistical analyses

Continuous variables were compared using either Student's *t*-test or one-way ANOVA with Dunnett's multiple comparison post hoc test. The frequency of categorical variables was compared using the $\chi^2$ test. We performed multivariate logistic regression analyses to evaluate the association of dementia with OH and the heart rate response. Clinical variables that were significant following a univariate analysis were included. The statistical analyses were performed using the JMP Version 14.2 software program (SAS Inc., Cary, NC, USA). A value of $P < 0.05$ was considered to be statistically significant.

## Result

### Baseline demographics and risk for dementia in PD

Table 1 shows the baseline demographics and the medical history of the enrolled PD patients. Of the 143 patients, 23 (16.1%) had dementia. The age, H-Y stage, cumulative years of education, and history of coronary artery disease (CAD) were significantly associated with dementia in PD patients. OH and SH were also associated with dementia, although there was no significant difference in the WMH scores on MRI between PD and PD with dementia (PDD). Each test related to cognitive function was significantly lower in the group with dementia than in the group without dementia. The LEDD was similar between the groups of PD and PDD. There were no significant differences in the use of anti-hypertensive medications, except for beta-blocker between PD and PDD.

The univariate analysis for the risk of dementia in PD showed a significant association with the age, H-Y stage, years of education, OH, SH and beta-blocker use (Table 2).

The multivariate odds ratios (ORs) for dementia in PD patients were significantly higher for the H-Y stage (Table 2, OR 2.6 per unit, 95% confidence interval [CI] 1.2–5.9, P<0.05), OH (Table 2, OR 8.9, 95% CI 1.6–49.0, P<0.05), and beta-blocker use (Table 2, OR 456.9, 95%

**Table 1. The comparison of the baseline characteristics between PD and PDD.**

|  | PD | PDD | P value |
|---|---|---|---|
| N = 143 | 120 (83.9%) | 23 (16.1%) |  |
| Age, y | 62.4±10.3 | 70.3±9.3 | <0.001 |
| Onset of age | 51.3±11.7 | 61.0±11.2 | <0.001 |
| Gender (F) | 71 (59.2%) | 9 (39.1%) | NS |
| Duration of disease, y | 11.2±6.7 | 9.3±6.0 | NS |
| H-Y stage | 2.8±0.8 | 3.5±0.9 | <0.001 |
| BMI | 21.8±3.6 | 21.2±3.8 | NS |
| Education, y | 13.6±2.2 | 12.1±3.1 | <0.01 |
| Hypertension, (%) | 24 (20.0%) | 8 (34.8%) | NS |
| Stroke, (%) | 2 (1.7%) | 2 (8.7%) | NS |
| Coronary artery disease, (%) | 1 (0.8%) | 2 (8.7%) | <0.05 |
| Peripheral artery disease, (%) | 0 | 0 | NS |
| Orthostatic hypotension, (%) | 59 (49.2%) | 19 (82.6%) | <0.01 |
| Supine hypertension, (%) | 14 (11.7%) | 8 (34.8%) | <0.01 |
| DWMH | 0.7±0.7 | 1.0±0.9 | NS |
| LEDD | 970.5±402.5 | 882.2±234.8 | NS |
| ACE-I/ARB | 13 (10.8%) | 1 (4.4%) | NS |
| Ca-blocker | 10 (8.3%) | 5 (21.7%) | NS |
| Beta-blocker | 1 (0.8%) | 3 (13.0%) | <0.01 |
| Diuretics | 2 (1.7%) | 0 | NS |
| HDS-R | 28.0±1.9 | 19.1±5.7 | <0.0001 |
| MMSE | 28.4±1.6 | 21.3±3.8 | <0.0001 |
| MoCA-J | 25.7±2.9 | 16.0±4.0 | <0.0001 |

CI 8.2–25304.2, P<0.01). In contrast, the ORs for dementia were significantly lower for years of education (Table 2, OR 0.7, 95% CI 0.5–0.9, P<0.05).

## The association between the orthostatic HR response and dementia in PD patients with OH

We divided OH into the two sub-types of neurogenic OH, in which the HR increase is < 15 beats per minute, and non-neurogenic OH in which the HR increase is ≥ 15 beats per minute. We then compared the differences in the risk for dementia between the no-OH group and each OH group (Table 3). The univariate ORs for dementia were significantly higher in the neurogenic OH group than the no-OH group (OR 7.3, 95% CI 2.2–24.6, P<0.01). This

**Table 2. The results of multivariable logistic regression analysis for the risk of dementia in PD.**

|  | Univariate | | | Multivariate | | |
|---|---|---|---|---|---|---|
|  | OR | 95% CI | P value | OR | 95% CI | P value |
| Age, per unit | 1.1 | 1.0–1.2 | <0.001 | 1.1 | 0.9–1.2 | NS |
| H-Y, per unit | 3.2 | 1.6–6.2 | <0.001 | 2.6 | 1.2–5.9 | <0.05 |
| Education, per unit | 0.8 | 0.6–0.9 | <0.01 | 0.7 | 0.5–0.9 | <0.05 |
| CAD | 11.3 | 0.9–130.7 | NS | - | - | - |
| OH | 4.9 | 1.6–15.3 | <0.01 | 8.9 | 1.6–49.0 | <0.05 |
| SH | 4.0 | 1.5–11.2 | <0.01 | 1.5 | 0.4–6.2 | NS |
| Beta-blocker use | 17.8 | 1.8–180.2 | <0.05 | 456.9 | 8.3–25304.2 | <0.01 |

**Table 3. The results of multivariable logistic regression analysis for the risk of dementia based on the type of orthostatic hypotension.**

|  | Univariate | | | Multivariate (model 1) | | | Multivariate (model 2) | | |
|---|---|---|---|---|---|---|---|---|---|
|  | HR | 95% CI | P | HR | 95% CI | P | HR | 95% CI | P |
| OH (-) | 1 (reference) | | | 1 (reference) | | | 1 (reference) | | |
| Non-neurogenic OH | 2.9 | 0.8–10.9 | NS | 3 | 0.7–12.4 | NS | 12.9 | 1.4–114.6 | <0.05 |
| Neurogenic OH | 7.3 | 2.2–24.6 | <0.01 | 5.3 | 1.5–18.8 | <0.05 | 11.5 | 1.6–85.1 | <0.05 |

difference remained significant after adjusting for the age, sex, H-Y stage, and years of education (model 1; OR 5.6, 95% CI 1.3–24.9, P<0.05) as well as for beta-blocker use (model 2; OR 11.5, 95% CI 1.6–85.1, P<0.05). We noted no significant differences in dementia between the non-neurogenic OH group and the no-OH group (OR 2.9, 95% CI 0.8–10.9, P = 0.12, model 1; OR 4.4, 95% CI 0.8–23.8, P = 0.08); however, there was a significant association after adjusting for beta-blocker use (model 2; OR 12.9, 95% CI 1.4–114.6, P<0.05).

## A comparison of the associated factors and cognitive impairment among patients without OH and with non-neurogenic or neurogenic OH

We used Dunnett's test to compare the factors associated with dementia and three independent cognitive scores among no-OH group and both OH subtypes (Table 4). The age and disease severity (H-Y score) were significantly higher in the neurogenic OH group than in the no-OH group. However, these associations were not observed between the non-neurogenic OH and no-OH groups. We also found respective significant differences in the presence of SH (4.6% vs. 10.5% vs. 37.5%, P<0.0001) and prevalence of dementia among patients with no-OH, non-neurogenic OH, and neurogenic OH (6.2% vs 15.8% vs 32.5%, P<0.01). The value of each cognitive score was lower in the neurogenic OH group than in the no-OH or no-neurogenic OH group. All cognitive scores differed significantly between the no-OH and neurogenic OH group, but we found no such association between no-OH and non-neurogenic OH groups

**Table 4. The comparison of the associated factors and cognitive impairment among patients without OH and with non-neurogenic or neurogenic OH.**

|  | No OH | Non-neurogenic OH | Neurogenic OH | P value |
|---|---|---|---|---|
| N = 143 | 65 | 38 | 40 |  |
| Age | 61.5±11.4 | 61.7±9.8 | 69.3±7.1[a] | <0.001 |
| Disease duration | 9.8±4.9 | 12.2±8.5 | 11.4±6.8 | NS |
| H-Y | 2.8±0.8 | 2.6±0.8 | 3.4±0.8[a] | <0.001 |
| Education (y) | 13.6±2.5 | 13.2±2.4 | 13.2±2.4 | NS |
| HT | 16 (24.6%) | 4 (10.5%) | 12 (30.0%) | NS |
| SH | 3 (4.6%) | 4 (10.5%) | 15 (37.5%) | <0.0001 |
| DWMH | 0.68±0.69 | 0.70±0.70 | 0.92±0.81 | NS |
| LED | 907.6±371.5 | 1013.0±346.6 | 981.6±425.6 | NS |
| Beta-blocker | 3 (4.6%) | 0 | 1 (2.5%) | NS |
| Dementia | 4 (6.2%) | 6 (15.8%) | 13 (32.5%) | <0.01 |
| HDS-R | 27.5±3.7 | 26.6±4.4 | 25.2±4.9[a] | <0.05 |
| MMSE | 28.1±2.8 | 27.4±3.0 | 25.7±4.1[a] | <0.01 |
| MoCA | 25.2±4.3 | 24.8±4.0 | 21.9±5.3[a] | <0.01 |
| ΔSBP | -2.0±11.0 | -31.3±14.9[a] | -31.5±21.2[a] | <0.001 |
| ΔDBP | 3.8±7.0 | -15.7±9.1[a] | -17.8±10.7[a] | <0.001 |

[a] significant differences compare to no OH groups.

(Table 4). While the blood pressure decrease was significantly lower in the both OH groups than in the no-OH group, there were no significant differences between the OH subtype groups (Table 4).

## Discussion

Although a previous systematic review article revealed the estimated prevalence of OH in PD patients to be 30.1% [17], the prevalence of OH was 54.5% in our cohort. The prevalence rate across studies has been reported to range from 9.6% to 64.9% and seems to be influenced by covariant factors, such as the age and disease duration [17]. As the disease duration (10.9 ± 6.6 years) at the assessment in the present study was relatively long, this might have increased the prevalence of OH in our cohort.

Although the association between OH and cognitive decline has been inconclusive, a recent meta-analysis of prospective cohort data showed the OH increased the risk of dementia, and this trend was preserved in two subtypes of dementia: Alzheimer's disease (adjusted pooled hazard ratio 1.175, 95% CI 1.022–1.351) and vascular dementia (adjusted pooled hazard ratio 1.403, 95% CI 1.042–1.889) [18].

Both OH and cognitive impairment may reflect common brain and peripheral neurodegeneration as well as its severity in PD [3, 19]. The anterior cingulate cortex has been proposed as an important site for cognitive and autonomic impairment [3]. The loss of integrity and atrophy in cingulate structural covariance networks has been associated with non-dopaminergic features, such as cognitive impairment and excessive daytime sleepiness [20]. The locus ceruleus, which is the sole source of noradrenaline, is also frequently affected in PD. Noradrenaline acts as a neuromodulator of multiple affected areas in the forebrain and influences the memory in attention or retrieve information [21]. Sommerauer et al. studied the association between the noradrenergic system and cognitive decline or OH in PD. They showed that a reduction in noradrenaline transporter on 11C-MeNER PET was associated with cognitive performance and OH [22].

The possible mechanisms underlying the cognitive impairment by OH are suspected to be multifactorial, but the most frequently proposed mechanism involves recurrent episodic brain hypoxia/ischemia. The cerebrovascular pathology and condition of ischemia/hypoxia are important pathogenic mechanisms underlying the development of neurodegeneration in dementia patients including α-synucleinopathies [23]. Interestingly, experimental models of brief cerebral blood flow reduction have promoted the aggregation of alpha-synuclein, which is associated with extensive neuronal cell death and large infarction [24]. These findings suggest that recurrent episodic brain ischemia/hypoxia induced by OH may indicate an increased risk of extensive aggregation of α-synuclein and thereby be associated with cognitive decline in PD patients.

Furthermore, brain hypoxia/ischemia by OH might induce development of small vessel disease in the brain. WMH on MRI is significantly more frequent in cases of OH, SH or both than in patients with neither among PD cases [25]. Our results also showed WMH was more severe in cases of neurogenic OH than in the no-OH or non-neurogenic OH patients, without statistical significance.

Cerebral microbleeds (CMBs) are also well-known markers of small vessel disease that can be detected on T2*-weighted gradient echo of MRI. CMBs are often observed in PD patients, and the presence of OH was found to be an independent risk factor for CMBs in our previous report [26]. CMBs were also significant greater risks for cognitive impairment in PD [27]. These small vessel diseases are thought to increase the neuroinflammation and deteriorate the pathology of α-synucleinopathies [23].

SH is often companied by OH in neurodegenerative disorders with autonomic dysfunction [19], and the neurogenic OH group showed a significantly higher prevalence of SH than no-OH and non-neurogenic OH in our cohort. Because CMBs are more abundant in cases of OH accompanied by SH than in OH without SH among PD patients [28], SH rather than OH may increase the susceptibility of end-organ damage and small vessel disease related to cognitive impairment in PD.

Our results showed that the absence of a heart rate increase was an important predictor of cognitive impairment. Walters et al. reported that the OH was associated with an increase in the long-term risk of dementia in a population-based prospective study, and the risk of dementia was markedly increased in those with OH who lacked a compensatory increase in the HR [29]. The lack of an HR increase may reflect the severity of autonomic dysfunction; however, we found no significant differences in the orthostatic drop in BP between neurogenic and non-neurogenic OH. As the risk of dementia was markedly increased in OH patients who lacked a compensatory increase in their HR, the formal assessment of the OH incorporation of the HR response is important for determining the severity of autonomic dysfunction and predicting cognitive impairment in PD patients.

Finally, our data showed that beta-blocker use was associated with PDD and non-neurogenic OH was also associated with dementia after adjusting for beta-blocker use. In this cohort study, beta-blocker use was found in only four patients, including two patients who used these agents for hypertension and another two patients who used them for tremor. The association between beta-blocker use and cognitive impairment has remained inconclusive [30, 31], and further larger scale studies will be needed to explore whether or not beta-blocker use is involved in the cognitive impairment in PD.

While we demonstrated an association between absence of heart rate increase and cognitive impairment in PD with OH, several limitations remain to be disclosed. Because of the retrospective nature study and the small patient population, further prospective studies in large population are warranted. Second, we did not review the data of syncope or unexplained fall in this study. Syncope and unexplained fall are important related symptoms of OH and are frequently observed in patients with dementia [32]. Further studies are also warranted to determine whether or not suffering from syncope or unexplained fall is related to cognitive impairment in PD.

In conclusion, this study showed that OH without an HR response was an important marker of cognitive impairment in PD. Further prospective studies should be conducted to clarify whether or not the HR response predicts cognitive decline.

## Author Contributions

**Conceptualization:** Ryota Tanaka.

**Data curation:** Ryota Tanaka, Kazuo Yamashiro.

**Formal analysis:** Ryota Tanaka.

**Investigation:** Ryota Tanaka, Kazuo Yamashiro, Takashi Ogawa, Genko Oyama, Kenya Nishioka, Atsushi Umemura, Yasushi Shimo.

**Project administration:** Ryota Tanaka, Kazuo Yamashiro.

**Supervision:** Nobutaka Hattori.

**Visualization:** Ryota Tanaka.

**Writing – original draft:** Ryota Tanaka.

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
