## [Decision Letter · Decision Letter 0]

17 Aug 2020

PONE-D-20-22293

The absence of orthostatic heart rate increase is associated with cognitive impairment in Parkinson’s disease

PLOS ONE

Dear Dr. TANAKA,

Thank you for submitting your manuscript to PLOS ONE. After careful consideration, we feel that it has merit but does not fully meet PLOS ONE’s publication criteria as it currently stands. Therefore, we invite you to submit a revised version of the manuscript that addresses the points raised during the review process.

We look forward to receiving your revised manuscript.

Kind regards,

Pasquale Abete

Academic Editor

PLOS ONE

Journal Requirements:

2.Thank you for stating the following in the Financial Disclosure section:

[R.T. reports personal fees from honoraria: not related to the current work: Takeda Pharmaceutical Co.,Ltd., Nippon Boehringer Ingelheim,Co.,Ltd, Dai-Nippon Sumitomo Pharma Co.,Ltd., Bayer Yakuhin, Ltd, Otsuka Pharmaceutical, Co.,Ltd, Pfizer Japan Inc., , DAIICHI SANKYO Co.,Ltd., Eisai Co.,Ltd., Bristol-Myers Squibb Co., Stryker Japan K.K., CSL Behring K.K. and Kowa Co.,Ltd.

K.Y. reports grants from Grants from Grant-in-Aid for Scientific Research, and personal fees from honoraria for work unrelated to the current study: Pfizer Inc,, Takeda Pharmaceutical Co.,Ltd., Dai-Nippon Sumitomo Pharma Co.,Ltd., Bayer Yakuhin, Ltd, Otsuka Pharmaceutical, Co.,Ltd., DAIICHI SANKYO Co.,Ltd., and Novartis Pharma K.K.

T.O. reports the following disclosures: None

K.N. reports the following disclosures: Grant from Japan Society for the Promotion of Science (JSPS) KAKENHI Grant and a research grant from Biogen Japan Ltd. not granted for the current study.

G.O. reports grants from Grants from Japan Society for the Promotion of Science; Grant-in-Aid for Young Scientists (B), honoraria from work unrelated to the current study: Medtronic, Boston Scientific, Otsuka Pharmaceutical, Sumitomo Dainippon Pharma, Novartis Pharma, MSD, Nihon Medi-Physics, FP Pharmaceutical Corporation, Kyowa Hakko Kirin, and Abbvie.

A.U. reports grants from Grants from Japan Society for the Promotion of Science; Grant-in-Aid for Scientific Research (C), Research Grant from Novartis Pharma K.K, consultancy: Boston Scientific Japan, and honoraria: Medtronic Japan Inc., Novartis Pharma K.K, Kyowa Hakko-Kirin Co.,Ltd., Takeda Pharmaceutical Co.,Ltd. And Abbvie, for work unrelated to the current study.

Y.S. reports grants from the Japan Society for the Promotion of Science, Grant-in-Aid for Scientific Research and speaker honoraria from Medtronic, Boston Scientific, Otsuka Pharmaceutical, Takeda Pharmaceutical CO, Sumitomo Dainippon Pharma, Novartis Pharma, MSD, FP Pharmaceutical Corporation, Kyowa Hakko Kirin, and AbbVie, Inc

N.H. reports grants from the Japan Society for the Promotion of Science (JSPS); grants from Ministry of Education Culture,Sports,Science and Technology Japan, grants from Health Labour Sciences Research Grant, grants from Japan Agency for Medical Research and Development (AMED), AbbVie GK, FP Pharma. Co, Dai-Nippon Sumitomo Pharma Co.,Ltd, Eisai Co.,Ltd., and received honoraria of as a speaker and advisory boards from Kyowa Kirin Co., Ltd., Dai-Nippon Sumitomo Pharma Co., Ltd, Takeda Pharmaceutical Co., Ltd., Otsuka Pharmaceutical, Co., Ltd, AbbVie GK, Eisai Co., Ltd., FP Pharma. Co, Mitsubishi Tanabe Pharma Co., Otsuka Pharmaceutical, Novartis Pharma, MSD, Nihon Medi-Physics, Bristol-Myers Squibb Company, Ono Pharmaceutical Co., Ltd., EA Pharma Co., Ltd, Asahi Kasei Medical Co., Ltd., and Chugai Pharma Manufacturing Co., Ltd.

Also he received honoraria from Biogen Japan Ltd for Subcontracting (Trial cases) and he has  the Equity stock (8%) of PARKINSON Laboratories Co. Ltd.]. 

We note that you received funding from multiple commercial sources.

Additional Editor Comments (if provided):

The authors retrospectively analyzed 147 cases of clinically diagnosed PD to determine the association between the absence of a heart rate response and cognitive impairment in PD with OH. Among the patients with OH, neurogenic OH was diagnosed in cases without a heart rate increase, while all other patients were diagnosed with nonneurogenicOH. Dementia was found in 25 of 147 PD cases (17%) in this cohort. The presence of OH was an independent risk factor for dementia in PD in addition to the disease severity and years of education. Neurogenic OH was significantly associated with dementia compared to the no OH group (harzard ratio [HR] 8.2, 95% confidence interval [CI] 2.5-26.9, P<0.001), an association that was preserved after adjusting for age, gender and other covariant factors. However, no such association was observed for non-neurogenic OH (HR 2.7, 95%CI 0.7-10.5, P=0.13). While the cognitive impairment was significantly worse in the neurogenic OH group than the no-OH group, the groups were otherwise similar. The blood pressure decrease was significantly lower in both OH groups than in the no-OH group, despite no significant differences between the OH groups.

The manuscript is interesting. However, I have a concern regarding the presence of syncope and/or unexplained falls in this sample. Please see and discuss Ungar A: Etiology of Syncope and Unexplained Falls in Elderly Adults with Dementia: Syncope and Dementia (SYD) Study. J Am Geriatr Soc. 2016 Aug;64(8):1567-73.

Reviewers' comments:

Reviewer's Responses to Questions

**Comments to the Author**

1. Is the manuscript technically sound, and do the data support the conclusions?

Reviewer #1: Yes

Reviewer #2: Partly

2. Has the statistical analysis been performed appropriately and rigorously? 

Reviewer #1: Yes

Reviewer #2: Yes

3. Have the authors made all data underlying the findings in their manuscript fully available?

Reviewer #1: Yes

Reviewer #2: Yes

4. Is the manuscript presented in an intelligible fashion and written in standard English?

Reviewer #1: Yes

Reviewer #2: Yes

5. Review Comments to the Author

Reviewer #1: The paper is of interest and the association between orthostatic hypotension (OH) , cognitive impairment and Parkinson’s disease (PD) is clinically relevant. The association between the heart rate response and cognitive impairment was retrospectively analyzed in 147 cases of clinically diagnosed PD to determine the association between the absence of a heart rate response and cognitive impairment in PD with OH. Among the patients with OH, neurogenic OH was diagnosed in caseswithout a heart rate increase, while all other patients were diagnosed with non-neurogenic OH. Dementia was found in 25 of 147 PD cases (17%). Neurogenic OH was significantly associatedwith dementia compared to the no OH group (harzard ratio [HR] 8.2, 95% confidenceinterval [CI] 2.5-26.9, P<0.001), an association that was preserved after adjusting forage, gender and other covariate. No association was observedfor non-neurogenic OH (HR 2.7, 95%CI 0.7-10.5, P=0.13). The manuscript is well written, data support conclusions.

Reviewer #2: The paper is interesting, resulting in a great impact on management of PD patients. The correlation of dementia in PD patients with neurogenic OH is a relevant finding to better understand evolution and prognosis of PD. However, some aspects should be more extensively argued to avoid confounding data.

Major revision

Materials and methods: In this study, drugs were not considered with exception of anti-hypotensive medications and levodopa. The influence of some classes of medications on postural response of arterial pressure is well known. In particular, there is no mention about beta-blockers. Patients with congestive heart failure were excluded from the study, but use of beta-blockers is also and often requested in hypertensive patients or in coronary artery disease (enrolled in this study). This class of medications could fade postural heart rate response leading to a wrong diagnosis of neurogenic OH.

The study should be improved by adding data on medications to avoid that the association of neurogenic OH with dementia in PD patients is considered only a winsome suggestion.

6. PLOS authors have the option to publish the peer review history of their article (what does this mean?). If published, this will include your full peer review and any attached files.

Reviewer #1: No

Reviewer #2: No

---

## [Author Response · Author response to Decision Letter 0]

14 Sep 2020

Response to reviewers,

Additional Editor Comments (if provided):

The authors retrospectively analyzed 147 cases of clinically diagnosed PD to determine the association between the absence of a heart rate response and cognitive impairment in PD with OH. Among the patients with OH, neurogenic OH was diagnosed in cases without a heart rate increase, while all other patients were diagnosed with nonneurogenicOH. Dementia was found in 25 of 147 PD cases (17%) in this cohort. The presence of OH was an independent risk factor for dementia in PD in addition to the disease severity and years of education. Neurogenic OH was significantly associated with dementia compared to the no OH group (harzard ratio [HR] 8.2, 95% confidence interval [CI] 2.5-26.9, P<0.001), an association that was preserved after adjusting for age, gender and other covariant factors. However, no such association was observed for non-neurogenic OH (HR 2.7, 95%CI 0.7-10.5, P=0.13). While the cognitive impairment was significantly worse in the neurogenic OH group than the no-OH group, the groups were otherwise similar. The blood pressure decrease was significantly lower in both OH groups than in the no-OH group, despite no significant differences between the OH groups.

The manuscript is interesting. However, I have a concern regarding the presence of syncope and/or unexplained falls in this sample. Please see and discuss Ungar A: Etiology of Syncope and Unexplained Falls in Elderly Adults with Dementia: Syncope and Dementia (SYD) Study. J Am Geriatr Soc. 2016 Aug;64(8):1567-73.

Response: 

We thank the editor for the important suggestion concerning the presence of syncope and unexplained fall in this study. Unfortunately, we did not collect information on syncope or unexplained fall in this study, so we could not analyze these issues. However, as the editor indicated, whether or not syncope and unexplained fall are risks for dementia and the relationship between orthostatic hypotension and syncope or unexplained fall in PD are clinically important issues. We have now mentioned this in the Discussion of the revised manuscript (page 18, line 267-271, page 198, line 272).

Reviewer #1: The paper is of interest and the association between orthostatic hypotension (OH) , cognitive impairment and Parkinson’s disease (PD) is clinically relevant. The association between the heart rate response and cognitive impairment was retrospectively analyzed in 147 cases of clinically diagnosed PD to determine the association between the absence of a heart rate response and cognitive impairment in PD with OH. Among the patients with OH, neurogenic OH was diagnosed in caseswithout a heart rate increase, while all other patients were diagnosed with non-neurogenic OH. Dementia was found in 25 of 147 PD cases (17%). Neurogenic OH was significantly associatedwith dementia compared to the no OH group (harzard ratio [HR] 8.2, 95% confidenceinterval [CI] 2.5-26.9, P<0.001), an association that was preserved after adjusting forage, gender and other covariate. No association was observedfor non-neurogenic OH (HR 2.7, 95%CI 0.7-10.5, P=0.13). The manuscript is well written, data support conclusions.

Rensponse: 

We thank the reviewer for this positive evaluation. In accordance with the comment from the academic editor and Reviewer 2, we modified the manuscript accordingly.

Reviewer #2: The paper is interesting, resulting in a great impact on management of PD patients. The correlation of dementia in PD patients with neurogenic OH is a relevant finding to better understand evolution and prognosis of PD. However, some aspects should be more extensively argued to avoid confounding data.

Major revision

Materials and methods: In this study, drugs were not considered with exception of anti-hypotensive medications and levodopa. The influence of some classes of medications on postural response of arterial pressure is well known. In particular, there is no mention about beta-blockers. Patients with congestive heart failure were excluded from the study, but use of beta-blockers is also and often requested in hypertensive patients or in coronary artery disease (enrolled in this study). This class of medications could fade postural heart rate response leading to a wrong diagnosis of neurogenic OH.

The study should be improved by adding data on medications to avoid that the association of neurogenic OH with dementia in PD patients is considered only a winsome suggestion.

Response:

First, we reviewed the data and found four patients had been treated with midodrine during the investigation. We apologize for our mistake, and those four patients have been excluded from this study. As the reviewer suggested, we reviewed the data again and added the information on anti-hypertensive medications such as angiotensin-converting-enzyme inhibitors (ACE-Is), angiotensin II receptor blockers (ARBs), calcium channel blockers (CCBs), beta blockers, and diuretics. There were no patients who were treated with ACE-Is. There were no significant differences in the prevalence of the use of any anti-hypertensive medications, except for beta-blockers, between PD and PDD (Table 1). Beta-blocker use was found in only four patients, including two patients who used these agents for hypertension and another two patients who used them for tremor. The basal results showed a similar trend to the original data; however, beta-blocker use was significantly associated with PDD (Table 2), and the ORs for dementia of non-neurogenic OH were significant after adjusting for beta-blocker use (Table 3). Because the number of patients prescribed beta-blockers in this study was very small, we were unable to draw any conclusion regarding whether or not beta-blocker use is a risk factors significantly associated with dementia in PD. Further, large-scale clinical studies will be needed to clarify this issue. 

We have now revised these results in the Abstract (page 3), Methods (page 7, line 87, line 90-93), and Results (page 10, line 140, line 146-147, page 11, line 151, line 154-157, page 12, line 167-174, page 13, line 185-187) sections and all tables. We also discussed this in the Discussion section (page 14, line 198, page 15, line 201, page 18, line 257-263, line 267-271, page 19, line 272).

---

## [Decision Letter · Decision Letter 1]

28 Sep 2020

The absence of orthostatic heart rate increase is associated with cognitive impairment in Parkinson’s disease

PONE-D-20-22293R1

Dear Dr. TANAKA,

We’re pleased to inform you that your manuscript has been judged scientifically suitable for publication and will be formally accepted for publication once it meets all outstanding technical requirements.

Kind regards,

Pasquale Abete

Academic Editor

PLOS ONE

Additional Editor Comments (optional):

No further comments.

Reviewers' comments:

Reviewer's Responses to Questions

**Comments to the Author**

1. If the authors have adequately addressed your comments raised in a previous round of review and you feel that this manuscript is now acceptable for publication, you may indicate that here to bypass the “Comments to the Author” section, enter your conflict of interest statement in the “Confidential to Editor” section, and submit your "Accept" recommendation.

Reviewer #1: All comments have been addressed

Reviewer #2: All comments have been addressed

2. Is the manuscript technically sound, and do the data support the conclusions?

Reviewer #1: Yes

Reviewer #2: Yes

3. Has the statistical analysis been performed appropriately and rigorously? 

Reviewer #1: Yes

Reviewer #2: Yes

4. Have the authors made all data underlying the findings in their manuscript fully available?

Reviewer #1: Yes

Reviewer #2: Yes

5. Is the manuscript presented in an intelligible fashion and written in standard English?

Reviewer #1: Yes

Reviewer #2: Yes

6. Review Comments to the Author

Reviewer #1: (No Response)

Reviewer #2: The authors have fully addressed my comments.

The correlation between the absence of heart rate response to postural change and dementia in patients affected by Parkinson disease should be an interesting subject to be more extensively treated in the future.

7. PLOS authors have the option to publish the peer review history of their article (what does this mean?). If published, this will include your full peer review and any attached files.

Reviewer #1: No

Reviewer #2: No

---

## [Editor Report · Acceptance letter]

2 Oct 2020

PONE-D-20-22293R1 

The absence of orthostatic heart rate increase is associated with cognitive impairment in Parkinson’s disease 

Dear Dr. Tanaka:

I'm pleased to inform you that your manuscript has been deemed suitable for publication in PLOS ONE. Congratulations! Your manuscript is now with our production department. 

Kind regards, 

on behalf of

Prof. Pasquale Abete 

Academic Editor

PLOS ONE